# Transcriptome Analysis of Goat Mammary Gland Tissue Reveals the Adaptive Strategies and Molecular Mechanisms of Lactation and Involution

**DOI:** 10.3390/ijms232214424

**Published:** 2022-11-20

**Authors:** Rong Xuan, Jianmin Wang, Xiaodong Zhao, Qing Li, Yanyan Wang, Shanfeng Du, Qingling Duan, Yanfei Guo, Zhibin Ji, Tianle Chao

**Affiliations:** Shandong Provincial Key Laboratory of Animal Biotechnology and Disease Control and Prevention, Shandong Agricultural University, 61 Daizong Street, Tai’an 271018, China

**Keywords:** goat, mammary gland, RNA-seq, involution, mammary cell remodeling, lactation

## Abstract

To understand how genes precisely regulate lactation physiological activity and the molecular genetic mechanisms underlying mammary gland involution, this study investigated the transcriptome characteristics of goat mammary gland tissues at the late gestation (LG), early lactation (EL), peak lactation (PL), late lactation (LL), dry period (DP), and involution (IN) stages. A total of 13,083 differentially expressed transcripts were identified by mutual comparison of mammary gland tissues at six developmental stages. Genes related to cell growth, apoptosis, immunity, nutrient transport, synthesis, and metabolism make adaptive transcriptional changes to meet the needs of mammary lactation. Notably, platelet derived growth factor receptor beta (*PDGFRB*) was screened as a hub gene of the mammary gland developmental network, which is highly expressed during the DP and IN. Overexpression of *PDGFRB* in vitro could slow down the G1/S phase arrest of goat mammary epithelial cell cycle and promote cell proliferation by regulating the PI3K/Akt signaling pathway. In addition, *PDGFRB* overexpression can also affect the expression of genes related to apoptosis, matrix metalloproteinase family, and vascular development, which is beneficial to the remodeling of mammary gland tissue during involution. These findings provide new insights into the molecular mechanisms involved in lactation and mammary gland involution.

## 1. Introduction

The mammary gland is an important exocrine gland located in the breast. It secretes milk for young animals and humans to provide rich protein, carbohydrates, lipids, and other nutrients [1]. Mammary gland development is dynamic and divided into three stages: embryonic, pubertal, and adult [2]. Mammary gland development in embryos and puberty occurs only slightly, while complete mammogenesis only takes place during pregnancy to become fully functional after parturition [3]. With the occurrence of the reproductive cycles, adult mammary tissue or cells will undergo repeat rounds of proliferation, functional differentiation, expansion, and degeneration. This indicates a high degree of mammary cell plasticity from gestation to involution stages [4,5]. Of course, this is inseparable from the regulation of hormones such as progesterone, estrogen, insulin, glucocorticoids, and prolactin [2,6]. In addition, the coordinated regulation of lactation physiological activity and mammary involution may also be mediated by large-scale changes in gene expression [7].

Currently, transcriptome sequencing technology has become an effective tool to reveal the gene expression characteristics of a mammary gland during lactation and involution. Transcriptome sequencing analysis of mammary tissues has been carried out in goats [8,9,10,11], sheep [7,12], cattle [13,14], pigs [15,16], mice [17,18] at different developmental stages. Interestingly, several studies also performed weighted gene co-expression network analysis on mammary gland tissue transcriptome sequencing data, and identified gene modules related to different lactation stages, milk yield, lactose content, and dairy fat, etc. [19,20,21]. To be sure, many genes and signaling pathways involved in the regulation of lactation and involution were discovered. These findings provide insights into the molecular genetic mechanisms involved in the regulation of lactation and mammary gland involution. However, the physiological processes of lactation and mammary involution are complex. On the one hand, the mammary gland has to make changes in physiological activity or tissue structure to adapt to lactation and involution [4]; on the other hand, mammary gland during lactation and dry periods is known to face challenges such as heat stress [22,23], nutrient distribution [24], and bacterial or viral infection [25], which are often accompanied by adaptive changes at the cellular, physiological, metabolic, and biochemical levels in the mammary gland [26]. Therefore, a more comprehensive and systematic transcriptome analysis of lactating and involutionary mammary gland tissues is necessary, especially for the economically valuable lactating livestock dairy goats.

To understand how genes precisely regulate goat mammary gland development, involution, mammary cell remodeling, physiological activities of lactation, and the molecular genetic mechanism behind various adaptive physiological activities, this study investigated the transcriptome characteristics of the mammary gland at the late gestation (LG), early lactation (EL), peak lactation (PL), late lactation (LL), dry period (DP), and involution (IN) stages (Figure 1). Furthermore, platelet-derived growth factor receptor beta *(PDGFRB*) was identified as a hub gene in the mammary gland developmental network by bioinformatics analysis during this study. The protein encoded by the *PDGFRB* gene is a cell surface tyrosine kinase receptor for members of the platelet-derived growth factor family [27]. *PDGFRB* plays an important role in the regulation of embryonic development, cell proliferation, survival, differentiation, chemotaxis, and migration [28,29,30]. However, the role of *PDGFRB* in goat mammary epithelial cells is unknown. Therefore, this study also analyzed the molecular mechanism of *PDGFRB* affecting the growth of goat mammary epithelial cells cultured in vitro. This work will provide theoretical support for the understanding of the regulation of lactation and mammary involution in the future.

## 2. Results

### 2.1. Basic Statistical Analysis Results of Sequencing Data

A total of 2.27 billion reads were obtained from the transcriptome sequencing data of goat mammary gland tissue, and the alignment rate of each sequencing library with the reference genome was ≥ 84% (Appendix A). A total of 63,825 transcripts were obtained, and after filtering according to the expression amount, 18,615 transcripts with FPKM ≥ 2 were obtained (Appendix A). The results of principal component analysis and cluster analysis between samples showed that, compared with before data filtering, the coefficient of variation of the first principal component and the second principal component decreased by 30.18% and 48.42%, respectively. Mammary samples at different developmental stages can be grouped according to principal component 1 and principal component 2 (Figure 2a,c). The results of sample relative log expression analysis showed that after normalization, 42 samples showed consistency in relative log expression analysis (Figure 2b,d). The results of the Venn diagram showed that 10,907 transcripts were expressed in all six developmental stages (FPKM ≥ 2), among which, the number of uniquely expressed transcripts was at most 1241 in the IN, and the lowest number of uniquely expressed transcripts in LL (Figure 2e). This reflects the temporal specificity of gene expression in the mammary gland. In addition, the highest number of transcripts was expressed in the DP and the least in the PL (Figure 2f). Analysis of the top 20 transcripts in each period (Appendix A) found that genes encoding milk proteins (*CSN1S1*, *CSN1S2*, *CSN2*, *CSN3*, and *LALBA*) were screened during lactation and non-lactation, but their expression levels were higher during lactation (Appendix A). Forty ribosome-associated transcripts were also screened, which were highly expressed in goat mammary glands at the LG (Appendix A). This is similar to the results of the overexpression of 18 ribosomal protein genes in bovine mammary gland tissue in late gestation [7]. This likely reflects increased levels of ribosome biosynthesis in preparation for high levels of milk protein synthesis.

### 2.2. Identification of Co-Expression Modules Associated with Different Stages of Mammary Gland Development

The WGCNA results show that the best soft threshold 6 is chosen to construct an approximately scale-free topological overlap matrix (Figure 3a,b). Modules were automatically assigned different colors to distinguish them from each other, while genes that were not clustered were grouped into module X (the grey module) (Figure 3c, Appendix A). Among them, 26,093 transcripts (FPKM ≥ 0.5) were assigned to 24 modules, of which the module X contained a total of 1899 genes, accounting for 7.3% of all analyzed genes (Figure 3c). The module L has the largest number of genes and the module F has the least number of genes. The correlations between different modules are shown through a cluster tree map and a heatmap, respectively (Figure 3e,f). The correlation analysis between modules and different developmental stages found that (Figure 4a), LG was significantly positively correlated with modules D and H (cor value ≥ 0.4, *p* value ≤ 0.008). EL was significantly positively correlated with module L (cor value = 0.42, *p* value = 0.006). PL was positively correlated with modules A, C, I, and J (cor value ≥ 0.42, *p* value < 0.05). LL was positively correlated with module B (cor value = 0.56, *p* value = 0.0001). DP was significantly positively correlated with modules K, T, U, and W (cor value ≥ 0.4, *p* value < 0.05). IN was significantly positively correlated with modules M, N, and O (cor value ≥ 0.55, *p* value < 0.05).

### 2.3. Analysis of Gene Function in Co-Expression Modules

A total of 12 gene modules (accounting for 52.17% of all modules) were significantly positively correlated with six stages (cor value ≥ 0.4, *p* value < 0.05, Figure 4a). Module L contained 4079 genes and was the module with the largest number of genes (Figure 4b), which was positively correlated with EL (cor value = 0.42, *p* value = 0.006). Six hub genes were identified from the module L (Figure 4c). GO analysis showed (Figure 4d) that GO terms related to gene transcription, such as RNA polymerase II transcription regulator complex and transcription regulator complex, were enriched. KEGG results showed (Figure 4e) that disease-related pathways such as mitophagy-animal, endocytosis, salmonella infection, and bacterial invasion of epithelial cells were significantly enriched. Interestingly, human T-cell leukemia virus 1 infection, pancreatic cancer, colorectal cancer, natural killer cell mediated cytotoxicity, and other diseases- and immune-related signaling pathways were significantly enriched to the modules K, U, and W, which are positively correlated with the DP (Appendix A). Similarly, pathways related to disease and immunity were also enriched into the M, N, and O modules correlated to the IN (Appendix A). This indicates that the mammary gland tissue is more susceptible to pathogenic microorganism invasion and disease in the early lactation, dry period, and involution period. At this time, the expression of immune-related genes in the mammary gland is more active to protect the health of mammary gland tissue in the early and dry periods of lactation. In module I correlated with the PL (Appendix A), GO terms such as long-chain fatty acid import into cell, cytoplasmic translation, and ribonucleoprotein complex biogenesis were found to be significantly enriched. In the module J (Appendix A), ribosomal subunit, prolactin signaling pathway, fatty acid biosynthesis, etc. were significantly enriched. In the module C (Appendix A), protein transmembrane transport, protein localization to endoplasmic reticulum, protein export, and cholesterol metabolism were enriched. The GO terms and KEGG signaling pathways related to the metabolism and transport were closely related to the milk synthesis in the mammary gland during the peak lactation period. Metabolic-related processes such as hormone metabolic process, regulation of lipolysis in adipocytes, and PPAR signaling pathway were found in the module B correlated to the LL (Appendix A). In the D module correlated to LG stage (Appendix A), cell cycle, insulin signaling pathway, longevity regulating pathway-multiple species, TGF-beta signaling pathway, and other pathways related to cell growth were enriched.

### 2.4. Differential Expression Analysis Results

There were 15 sets of differential comparisons between mammary glands at six different developmental stages, and a total of 13,083 differentially expressed transcripts were identified (Figure 5a and Appendix A). Among them, the highest number of differentially expressed transcripts (DETs) were found in EL vs. LG and the lowest in EL vs. LL. In addition, the Upset plot indicated the distribution of DETs between different comparison groups (Figure 5b), in which 55 transcripts were differentially expressed among all comparison groups, and 843 DETs were only present in PL vs. DP. The heatmap showed that 13,083 transcripts had changes in expression over the six periods (Figure 5c). According to the cluster analysis of the expression levels of DETs in different periods (Figure 5d), six clusters of DETs with different expression characteristics were obtained. Among them, the maximum number of DETs in cluster 1 is 3412, and the number of DETs in cluster 4 is the least, which is 1063 (Appendix A). These results suggest that gene transcript levels are altered during lactation and involution in goat mammary tissue from LG to IN stages, reflecting the stage-specificity of gene expression in mammary gland tissue.

### 2.5. GO and KEGG Pathway Enrichment Analysis of DETs

Cluster 1 showed DETs that were highly expressed in late gestation (Figure 5d). GO results showed (Appendix A) that terms related to protein synthesis and modification, such as ribonucleoprotein complex biogenesis, cytoplasmic translation, ribosome biogenesis, and regulation of protein ubiquitination, were significantly enriched. In addition, terms related to cell proliferation and division, such as RNA splicing, mitotic nuclear division, and mitotic nuclear division, were enriched. KEGG results showed that ribosome, ribosome biogenesis in eukaryotes, spliceosome, cell cycle, DNA replication, and other signaling pathways related to cell growth were enriched. This is closely related to the growth of mammary gland tissue in late gestation. Cluster 2 results show transcripts that are highly expressed during the EL period. Longevity regulating pathway, AMPK signaling pathway, Autophagy-animal, mTOR signaling pathway, and other signaling pathways related to cell growth were enriched. Cluster 3 and cluster 4 displayed transcripts that were highly expressed during the dry period. Immune-related terms such as mononuclear cell differentiation, T cell activation, lymphocyte differentiation, positive regulation of interleukin-2 production, and regulation of mononuclear cell migration were enriched, indicating that genes related to mammary gland immunity were significantly up-regulated during the dry period. On the one hand, it removes senescent and apoptotic cells; on the other hand, it improves the protection of mammary gland health during the dry period. Cluster 5 displays transcripts that are highly expressed during peak lactation. Terms related to substance synthesis and metabolism, such as phospholipid biosynthetic process, phospholipid metabolic process, and alpha-amino acid metabolic process, are enriched. This is related to the mammary gland cell’s lactation activity and exuberant cell metabolism during the peak lactation period. Cluster 6 displays transcripts that are highly expressed in involutionary mammary tissue. Cell growth, regulation of cell growth, developmental cell growth, PI3K-Akt signaling pathway, etc. cell growth-related terms were enriched. In addition, cell-substrate adhesion, extracellular matrix organization, and adherens junction were enriched. This reflects that mammary gland cells undergo extensive cellular remodeling in addition to intense apoptosis during mammary gland involution, thereby preparing for a new round of lactation.

### 2.6. Substance Synthesis, Transport and Metabolism Related Gene Analysis

A total of 77 genes were screened to be related to substance synthesis and metabolism, and seven of them were identified as network hub genes (Appendix A). These genes are involved in amino acid (cellular amino acid metabolic process, cellular amino acid biosynthetic process, and cellular amino acid catabolic process), carbohydrate (carboxylic acid catabolic process, carboxylic acid biosynthetic process, and glucose metabolic process), lipid (fatty acid metabolic process), long-chain fatty acid transport, and lipid import into the cell) and other metabolic processes (Appendix A). The expression levels of hub genes *IDH1*, *PDHB*, *PC*, *TKT*, *SDHA*, and *ALDH18A1* were higher in lactation than in the dry period and involution. *IDH1*, *PDHB*, *SDHA*, *PC*, and *TKT*, all participate in the carbon metabolism signaling pathway. In addition, *ALDH18A1*, *IDH1*, and *TKT* are involved in the process of biosynthesis of amino acids. Notably, the PPAR signaling pathway related to lipid metabolism was also significantly enriched. Compared with the dry period and involution period, the member genes *CD36*, *PLIN2*, *LPL*, *FABP3*, *DBI*, etc., in the PPAR signaling pathway were also highly expressed in the lactation period (Appendix A). The above shows that the energy and material metabolism levels of the mammary gland decrease from lactation to the dry period, and the genes related to protein, lipid, and carbohydrates are also adjusted at the transcriptional level to adapt to this transition process.

### 2.7. Analysis of Immune Function Genes Expressed in Mammary Glands

A total of 60 immune-related genes were screened (Appendix A), and five genes were identified as hub genes. These genes are involved in innate immunity (leukocyte proliferation, leukocyte cell-cell adhesion, natural killer cell mediated cytotoxicity) and adaptive immune processes (regulation of adaptive immune response, lymphocyte proliferation, and immune receptors built from immunoglobulin superfamily domains) (Appendix A). The hub genes *CD86*, *CCL5*, *PTPRC*, *CCL2*, and *IL1B* were highly expressed during the dry period (Appendix A). *CD86* is involved in mononuclear cell differentiation. *CCL5* is involved in the regulation of leukocyte proliferation. *CCL2* can regulate positive regulation of leukocyte activation. IL1B and PTPRC are involved in the regulation of lymphocyte differentiation. In addition, genes *CD74*, *RAC2*, *CXCL10*, *FCER1G*, *ITGB2*, etc. related to neutrophil migration were highly expressed during the dry period. The genes associated with macrophage activation *CD74*, *PTPRC*, *ITGAM*, *ITGB2*, *TYROBP*, *TNF*, *AIF1*, and *THBS1* were highly expressed during the dry period. Mammary gland tissue is known to undergo apoptosis and involution during the dry period, suggesting a role for neutrophils and macrophages in the clearance of casein micelles, lipid droplets, and cellular debris during the dry period.

### 2.8. Analysis of Genes Related to Cell Apoptosis and Autophagy

Protein-protein interaction analysis of 45 differentially expressed genes related to apoptosis and autophagy revealed 15 hub genes in the PPI network (Appendix A). KEGG results showed that 35 genes were involved in the apoptosis signaling pathway, and 17 genes were related to the autophagy pathway (Appendix A). GO functional annotation analysis showed that GO terms such as extrinsic apoptotic signaling pathway, cellular response to external stimulus, and positive regulation of cysteine-type endopeptidase activity involved in the apoptotic process were also enriched. The hub gene expression analysis showed (Appendix A), *TP53*, *BCL2A1*, *TNF*, *FAS*, *BAD*, *TRAF2* (*XM_018044495.1*), *SQSTM1*, and *BAX* were highly expressed in the DP stage. *MAPK1* and *FOS* are highly expressed in the IN. *CTSL*, *PMAIP1*, *TRAF2* (*XM_018044493.1*), *CTSB*, *CTSD*, and *TRADD* are highly expressed in DP and IN stages. These findings support that expression of genes involved in the physiological process of apoptosis and autophagy are altered at the transcriptional level in degenerative mammary tissue.

### 2.9. Analysis of Genes Related to Mammary Gland Development and Cell Growth

A total of 226 genes related to mammary gland development were screened, of which 9 genes were identified as hub genes (Appendix A). These genes are involved in physiological processes such as extracellular matrix organization, tissue remodeling, gland development, and morphogenesis of a branching epithelium (Appendix A). In addition, cellular response to growth hormone stimulus, prolactin signaling pathway, Insulin signaling pathway, and other hormone-related signaling pathways were also significantly enriched. Cytokine-related terms such as response to fibroblast growth factor, response to transforming growth factor beta, and vascular endothelial growth factor receptor signaling pathway were also enriched. It is worth noting that the PI3K-Akt signaling pathway is the most significant signaling pathway, which contains 77 genes. A total of 146 transcripts (from 77 genes) were clustered into five clusters according to their expression levels (Appendix A). Among them, 61 transcripts were highly expressed in DP, and 60 transcripts were highly expressed in IN. Some 17 transcripts were highly expressed during the LG period. The PPI results showed that five genes were identified as hub genes of the PI3K-Akt signaling pathway (Figure 6c). This study found that *PDGFRB* was highly expressed in mammary gland tissue during DP and IN stages and was located upstream of the PI3K-Akt signaling pathway (Figure 6d and Appendix A). However, the roles and mechanisms of *PDGFRB* in controlling mammary epithelial cell growth are unclear. Therefore, the role of the PDGFRB in mammary cells in vitro was deeply explored.

### 2.10. PDGFRB Overexpression In Vitro Promotes the Proliferation of Mammary Epithelial Cells by Activating PI3K-Akt Signaling Pathway

To verify the effect of *PDGFRB* on the proliferation of mammary epithelial cells, we interfered with or overexpressed PDGFRB protein in mammary epithelial cells cultured in vitro, and measured the cell viability of cells transfected 48 h by EdU (5-Ethynyl-2′-deoxyuridine) staining. Western blotting results showed (Figure 6e) that the expression level of PDGFRB in the interference group was significantly lower than that in the control group (48.22 ± 1.14% lower; *p* < 0.01), and the expression level of PDGFRB in the overexpression group was significantly higher than that in the control group (59.36 ± 13.92% higher; *p* < 0.01). EdU results showed significant differences in cell proliferation activity 48 h after cell transfection (Figure 7a). At 48 h, compared with the control group, the cell proliferation activity in the interference group was significantly decreased by 67.74 ± 10.03%, while that in the overexpression group was significantly increased by 107.92 ± 5.78% (*p* < 0.01). In addition, Western blotting results showed (Figure 7) that compared with the control group, interference with PDGFRB reduced the ratio of p-PI3K/PI3K and p-Akt/Akt, and increased the expression of p53 and p21. The overexpression of PDGFRB increased the ratio of p-PI3K/PI3K and p-Akt/Akt, and decreased the protein expression levels of p53 and p21. These data suggest that PDGFRB overexpression in vitro promotes the proliferation of mammary epithelial cells by activating the PI3K-Akt signaling pathway.

### 2.11. Overexpression of PDGFRB Inhibits Apoptosis of Mammary Epithelial Cells and Reduces Cell Cycle Arrest in G1/S Phase

Cell cycle analysis showed (Figure 8a) that 48 h after transfection, compared with the control group, interference with PDGFRB increased the number of cells in G 1/S phase by 25.31 ± 5.62%, decreased the number of cells in G 2/M phase by 16.09 ± 3.64%, and decreased the expression of CCND1. In contrast, PDGFRB overexpression increased CCND1 expression and caused more cells to enter the G 2/M phase. Apoptosis detection showed that (Figure 8c), the apoptosis rate of the PDGFRB interference group was 49.96% ± 8.71% higher than that of the control group. In contrast, the apoptosis rate of the overexpression group was 47.28 ± 3.37% lower than that of the control group. Western blotting results showed that the ratio of Bcl-2/Bax in the interference group was significantly lower than that in the control group (*p* < 0.01). In contrast, the Bcl-2/Bax ratio in the overexpression group was significantly higher than that in the control group (*p* < 0.01) (Figure 8e). Overexpression of PDGFRB also effectively reduced the expression of caspase 3 (Figure 8f). In conclusion, PDGFRB overexpression activates the PI3K/Akt signaling pathway, inhibits mammary epithelial cell apoptosis, changes the cell cycle by upregulating CCND1, reduces the number of arrested cells in the G1/S phase, and is beneficial to mammary epithelial cell proliferation.

### 2.12. Effects of PDGFRB on Genes Related to Lactation Function and Mammary Gland Involution

Compared with the control group (Figure 9), overexpression of PDGFRB significantly increased the expression of *MMP2*, *MMP19*, *AKT1*, *STAT1*, *IRF6*, *JAK2*, *VEGFA*, and *IRS2* at transcriptional level in cultured mammary epithelial cells, while significantly decreased *IGFBP5* expression. JAK-STAT signaling is known to play an important role in the initiation of mammary gland involution. Furthermore, matrix metalloproteinases (MMPs) constitute a family of proteases that play a major role in mammary gland involution. Specifically, these enzymes are involved in the degradation of the extracellular matrix, basement membrane, or both. Therefore, it is speculated that *PDGFRB* may be involved in the process of inducing mammary gland involution. However, interference or overexpression of *PDGFRB* in mammary epithelial cells had no effects on encoding milk proteins (*CSN1S1*, *CSN1S2*, *CSN2*, *CSN3*, and *LALBA*), cell growth (*FGFR2*, *MMP9*, *TGFB3*, and *MAPK1*), and hormone receptor-related genes (*ESR1* and *PRLR*).

## 3. Discussion

The mammary gland is a dynamic gland capable of undergoing repeated cycles of growth, differentiation, and involution coordinated by reproductive states [31]. Studying the molecular mechanisms of mammary gland development and function will provide fundamental insights into tissue remodeling and a better understanding of milk production and mammary diseases [7]. This is important for livestock production systems and human health. With the rapid development of sequencing technology, more and more transcriptome studies related to mammary gland tissue have been reported in goats [10,32,33,34,35], sheep [7,36,37], cows [14,38,39], mice [17,40], and other animals. However, a complete mammary transcriptome analysis covering all six stages from late gestation to involution has rarely been reported in goats. In this study, we constructed gene co-expression networks associated with six different stages of the lactation cycle in the goat mammary gland (Figure 3). Through differential expression analysis (Figure 5), 13,083 differentially expressed transcripts in six stages were identified, and gene networks related to the lactation stage, cell growth, substance metabolism, transport, and immunity were found. This indicates that the expression of genes at the transcriptional level in the mammary gland has undergone large-scale changes from late gestation to involution, thus coordinating to adapt to various structural and physiological changes in the mammary gland tissue.

Genes with similar expression patterns may be functionally related, and identifying genes with similar expression can shed more light on their potential functions [41]. WGCNA is an effective data mining method, which modularizes large data sets and obtains highly biologically significant co-expression modules based on similar gene expression patterns [41]. In recent years, WGCNA has been applied to reveal lactation-related genes in Holstein cows, and specific modules related to lactation stage, milk production, lactose, fat, and protein content have been found [19,20]. In addition, modules related to bovine mastitis were also identified, among which 250 genes highly associated with mastitis were identified [42]. However, there are few reports on goat mammary gland using WGCNA method. A total of 42 mammary gland tissue samples from six different developmental stages were analyzed in this study. Twelve modules related to the lactation stage were identified (Figure 4). Functional analysis of genes in the module showed that long-chain fatty acid import into cell, cytoplasmic translation, ribonucleoprotein complex biogenesis, and other pathways involved in fatty acid, carbohydrate, protein synthesis, and transport were enriched through lactation (EL, PL, and LL). This is consistent with previous results of mammary tissue analysis before, during, and after the peak lactation [19]. These results indicate that during lactation, cellular metabolic activity and nutrient transport increase to provide substrates for the synthesis of milk components. In addition, hormone and cell growth-related signaling pathways such as growth hormone synthesis, secretion and action, ErbB signaling pathway, cell cycle, and insulin signaling pathway were found in the modules related to late gestation and early lactation (Appendix A). Because physiologically, late gestation is characterized by extensive structural remodeling, including expansion of the lobule-alveolar network, and functional differentiation of alveolar cells into secretory cells in preparation for milk production during lactation [43]. Notably, terms related to immunity, apoptosis, and tissue remodeling were found in the dry period and involution stage (Appendix A). A known feature of active mammary gland involution is the recruitment of immune cells. In addition, the total number of white blood cells in milk increased rapidly during the first 3 days of degeneration and remained elevated until calving [44]. In this study, the regulation of leukocyte differentiation, the physiological process of mononuclear cell differentiation, was enriched in the module K (Appendix A). This proves that immune genes are involved in the process of the goat mammary gland involution from the level of gene transcription.

The maintenance of normal mammary gland development and lactation must depend on the coordinated regulation between cell survival and death. Autophagy and apoptosis control the turnover of intracellular organelles and proteins, and the turnover of cells in organisms respectively, which are critical for maintaining mammary tissue homeostasis [45,46]. Apoptosis is involved in lumen formation and acinar morphogenesis, replacement of cells during lactation, when mammary epithelial cells exhibit high secretory activity, and in the involution period of the mammary gland, and is involved in virtually every stage of mammary gland development. This study found that apoptosis-related signaling pathways such as the extrinsic apoptotic signaling pathway, intrinsic apoptotic signaling pathway in response to endoplasmic reticulum stress, and TNF signaling pathway were significantly enriched (Appendix A). In addition, apoptosis-inducing genes *TP53*, *BAX*, *TNF*, *PMAIP1*, *FAS*, and *GADD45A* were highly expressed in the DP, and *BAK1*, *SPTAN1*, and *RAF1* were highly expressed in the IN (Appendix A). This is closely related to the apoptosis of mammary gland cells during dry periods and involution. Similarly, apoptosis-related signaling pathways (e.g., JAK-STAT signaling pathway, IGF–IGF-Binding Protein System, and metalloprotease system) and genes (eg., *IGFBP5*, *STAT3*, *TGFB1*, and *MMP9*) are also activated in mice mammary glands during involution [34,47]. However, there are differences in the speed and degree of mammary gland involution across species. Among them, the number and degree of apoptosis of mammary cells in rodents were significantly more severe than that in cows. This discrepancy relative to rodents can be partly explained by the fact that cows typically conceive during dry periods. Therefore, hormones and local signals that act on mammary gland development during pregnancy may counteract those that act during involution in cows [4]. Interestingly, extensive apoptosis occurred in goat mammary glands at the late lactation and dry period by the TUNEL staining assay. This is consistent with the findings of this study that pro-apoptotic genes are up-regulated during dry and involution periods. On the one hand, this indicates that the goat mammary gland undergoes apoptosis during involution; on the other hand, it also reflects the difference in the mammary involution process between goats and cows. In addition to apoptosis during involution, mammary cells die due to lysosomal autophagy [48]. It has been shown that more than 20 traditional markers of lysosomal activity are up-regulated within 24 h after forced weaning, which suggests that autophagy is induced in the early stages of involution [49]. Furthermore, recent studies have demonstrated that apoptosis in degenerated bovine mammary cells is accompanied by an increase in the intensity of autophagy and that TGFB1 can induce apoptosis and autophagy in bovine mammary epithelial cells [50]. This study also detected a high expression of autophagy-related genes in goat mammary tissue during involution (Appendix A). This reflects the role of autophagy in mammary gland degeneration and remodeling. It regulates well the balance between protein degradation and organelle renewal in mammary cells [51,52].

The mammary gland is a vigorous gland that undergoes dramatic physiological adaptations during the transition from lactation to dry period, and then to lactation. On the one hand, this is inseparable from the precise regulation of hormones [53]; on the other hand, genes play an important role in regulating the differentiation of mammary epithelial cells into a secretory phenotype. Among them, genes related to substance metabolism have made adaptive changes. Carbon metabolism is the main source of energy required by organisms and provides precursors for other metabolisms in the body. This study found that *FBP1* (Fructose-Bisphosphatase 1), a gluconeogenesis-regulating enzyme, was highly expressed during lactation (Appendix A). Glucose homeostasis is known to be mutually controlled by the catabolic glycolytic and anabolic gluconeogenesis pathways, and *FBP1* acts as a rate-limiting enzyme in gluconeogenesis and can regulate cell growth and metabolism in mammary cells [54]. In addition, this study found that glutamic-oxaloacetic transaminase 1 (*GOT1*) was highly expressed at late lactation, which may be related to the regulation of mammary cell proliferation and apoptosis. There is already evidence that inhibition of *GOT1* promotes pancreatic cancer cell death via ferroptosis [55]. However, it has also been found that, relative to quiescent cells, proliferating cells catabolize more glutamate by transaminases, combining non-essential amino acid (NEAA) synthesis with α-ketoglutarate production and tricarboxylic acid (TCA) cycle replenishment, so the highly expressed *GOT1* can promote the proliferation of mammary epithelial cells [56]. In addition, *GOT1* regulates cellular metabolism to meet nutritional requirements by coordinating carbohydrate and amino acid utilization [57]. These all indicate the diversity of *GOT1* functions and the complexity of the mechanism of action.

Milk protein contains all the essential amino acids and other amino acids for human growth and development, and the mammary gland is the factory for milk protein synthesis and processing. This study found that the expression of genes related to amino acid transport and metabolism were altered in goat mammary tissue at different developmental stages. In addition to the high expression of genes encoding milk proteins during lactation, this study also found the expression levels of solute carrier family members (*SLC7A7*, *SLC16A2*, *SLC25A13*, *SLC25A12*, and *SLC39A8*), *CLN3*, *SERINC5*, *CTNS*, and other genes involved in amino acid transport changed at different stages (Appendix A). Our findings are similar to the reports that the expression of transporters of the solute carrier (SLC-) superfamily varies with the lactation stage of the mammary gland [58,59,60]. The solute carrier (SLC) family of membrane transporters includes more than 400 members divided into 66 families. They are gatekeepers to all cells and organelles, controlling the uptake and efflux of key compounds such as sugars, amino acids, nucleotides, inorganic ions, and drugs [61]. These reflect the importance of members of the solute carrier family in the material transport process in the mammary gland. In addition, *CLN3* was highly expressed in early and late lactation. *CLN3* is known to encode proteins that can be involved in the lysosomal function, mediate microtubule-dependent anterograde transport connecting the golgi network, endosomes, autophagosomes, lysosomes, and the plasma membrane, and is involved in a variety of cellular processes such as regulation lysosomal pH, lysosomal protein degradation, receptor-mediated endocytosis, autophagy, etc. [62,63]. Therefore, it is speculated that *CLN3* not only regulates the transport and synthesis of substances, but may also play a regulatory role in the apoptosis of mammary gland cells and the maintenance of cell homeostasis at the end of lactation. Members of the aldehyde dehydrogenase family (*ALDH18A1*, *ALDH4A1*, *ALDH5A1*, and *ALDH1A1*) are highly expressed during lactation (Appendix A) and may play an important role in maintaining normal amino acid metabolism in mammary cells. *ALDH18A1* is known to encode a bifunctional ATP and NADPH-dependent mitochondrial enzyme with γ-glutamyl kinase and γ-glutamyl phosphate reductase activities [64]. *ALDH4A1* can convert pyrroline-5-carboxylate to glutamate [65]. In addition, normal mammary cells with a high ALDH activity are known to have the broadest lineage differentiation potential and the highest ability to produce outgrowths in vivo compared to other mammary epithelial cell populations [66]. Aldehyde dehydrogenase activity is also a biomarker of primitive normal human breast luminal cells [67]. Therefore, the high expression of aldehyde dehydrogenase during lactation may be closely related to the strong mammary cell viability during lactation, thus adapting to the needs of mammary gland development and to the lactation function [68,69]. In conclusion, genes related to amino acid metabolism were altered at the transcriptional level at different developmental stages to meet the needs of mammary gland development and lactation.

The metabolism and growth of fat in the mammary gland are also very characteristic. The mammary glands of lactating mice synthesize and secrete milk fat equivalent to their entire body weight during a 20-day lactation cycle, making them one of the most active lipid-synthesizing organs known [70]. Additionally, butterfat is arguably one of the most complex fats in nature and varies from animal to animal. This study found that the PPAR signaling pathway regulates lipid metabolism and milk fat synthesis in goat mammary glands (Appendix A). It has been reported that the PPAR signaling pathway regulates lipid synthesis and metabolism in the mammary gland in sheep [7], cattle [39], pigs [15], and other animals. In this study, it was found that *PLIN2*, a member of this pathway, was highly expressed in EL and PL (Appendix A). The protein encoded by this gene belongs to the perilipin family, members of which encapsulate lipid storage droplets within cells. Most of the lipids produced during lactation are secreted into milk through a new process of membrane wrapping of cytoplasmic lipid droplets (CLDs), thus presumably playing a vital role in the formation and secretion of milk fat [71]. *PLIN2* transcripts increase more than 30-fold in milk-secreting cells as mammary glands differentiate into secretory glands during pregnancy [72]. In fact, microarray analysis revealed that the transcript of *PLIN2* is one of the most abundant in the lactating mouse mammary gland, comparable to other milk-secreting proteins such as casein [73]. This is consistent with our findings in the mammary glands of goats at different developmental stages. *LPL*, *FABP3*, and *FABP5* were also enriched in the PPAR signaling pathway, all of which were up-regulated during lactation. *LPL* encodes lipoprotein lipase, and in the mammary gland, very low-density lipoprotein (VLDL) or chylomicrons are anchored to the mammary endothelium by lipoprotein lipase (*LPL*), which then hydrolyzes TAGs in the lipoprotein core to release fatty acids [74]. *LPL* is more active in mammary glands compared to other tissues, possibly due to its high mRNA abundance [4,75]. In addition, *LPL* was found to be up-regulated during lactation in both bovine and mouse mammary gland tissues [75]. This suggests an important role of *LPL* in maintaining milk synthesis. *FABP3* and *FABP5* are members of the intracellular fatty acid-binding protein (FABPs) family and are thought to be involved in lipid metabolism through the binding and intracellular transport of long-chain fatty acids. mRNAs for all FABP isoforms are present in bovine mammary gland tissue, and their mRNA abundance is upregulated during lactation. Among them, the transcripts of *FABP4* and *FABP5* were up-regulated in lower amounts during lactation than those of *FABP3* [76]. In addition to the above genes, *CD36*, *DBI*, *OLR1*, *HMGCS1*, *ACSL1*, and *ME3* are also involved in the regulation of the PPAR signaling pathway. PPAR regulates the genes mentioned above, enriching the role of PPAR signaling pathway in goat mammary lipid metabolism.

It is well known that the number and activity of secretory epithelial cells affect milk production [77], and in particular, the stage of mammary tissue establishment that occurs during pregnancy is critical for subsequent lactation performance [7]. This study found that the cell cycle, p53 signaling pathway, rRNA metabolic process, and other cell growth-related terms are enriched in the late gestation stage (Appendix A). It is known that p53 signaling is involved in coordinating cellular responses to different types of stress (such as DNA damage and lack of Oxygen), downstream signaling in this pathway leads to apoptosis, senescence, and cell cycle arrest [78]. Furthermore, appropriate levels of p53 activity are important in regulating mammary duct morphogenesis, in part by regulating the IGF-1 pathway [79]. During the involution stage, terms such as extracellular structure organization, extracellular matrix organization, and cell-substrate adhesion are found. The components of the extracellular matrix (ECM) are known to link together to form structurally stable composites that contribute to the mechanical properties of the tissue. At the same time, the ECM is also a repository for growth factors and bioactive molecules. The mammary gland is a highly dynamic entity with changes in composition and structure of the ECM with pregnancy, lactation, and involution. Furthermore, it determines and controls the most basic behaviors and characteristics of cells such as proliferation, adhesion, migration, polarity, differentiation, and apoptosis [80,81]. In this study, it was found at the molecular level that ECM-related genes were up-regulated during involution, regulating the mammary gland involution process. In addition, the PI3K-Akt signaling pathway, axon guidance, focal adhesion, and other pathways related to cell growth were also enriched in the involution stage. These provide insights into the dynamic changes of genes during mammary gland development and involution.

Interestingly, 77 genes related to mammary gland development screened in this study were enriched in the PI3K-Akt signaling pathway (Figure 6). This pathway is involved in mammary gland cell metabolism [82], growth [83], involution [8], and immunity [84], as has been widely reported. *PDFGRB* was screened as the hub gene of the PI3K-Akt signaling pathway, which was highly expressed in the dry period and involution stage. *PDGFRB* is known to be abundantly expressed in normal stromal fibroblasts and advanced breast cancer cells [85,86]. Lineage tracing studies have established *PDGFRB* as a marker for adipocyte progenitors [87]. Inhibition of PDGFR signaling interferes with epithelial-mesenchymal transition (EMT) and leads to apoptosis in mouse and human breast cancer cell lines [88]. Extensive apoptosis of mammary epithelial cells during mammary involution, re-establishment of adipose and connective tissue as major components of non-secretory mammary glands, and increased fibrillar collagen deposition during involution. Therefore, it is speculated that these physiological processes may be related to *PDGFRB*. Indeed, it was found in this study (Figure 7 and Figure 8) that overexpression of *PDGFRB* can promote the proliferation of goat mammary epithelial cells, whereas inhibition of *PDGFRB* induces cell cycle G1/S arrest in goat mammary epithelial cells, which in turn induces apoptosis. Similarly, overexpression of *PDGFRB* in the mouse mammary gland causes abnormal proliferation of normal mouse mammary epithelial cells [89]. Not only that, the overexpression, point mutation, deletion, and translocation of *PDGFRB* play an important role in tumorigenesis and maintenance of malignant phenotype [90]. Taken together, these results suggest that *PDGFRB* is an important factor regulating mammary cell growth and that the overexpression of *PDGFRB* during involution may promote mammary gland remodeling.

The overexpression of *PDGFRB* not only regulates the members of the PI3K-Akt signaling pathway but also significantly up-regulates the expression levels of *MMP2* and *MMP19* transcription levels and down-regulates the expression of *IGFBP5* in mammary epithelial cells cultured in vitro (Figure 9). Transgenic mice expressing *IGFBP5* on a mammary gland-specific promoter result in impaired mammary gland development, including inhibition of IGF signaling and Bcl-2 family members [91]. This study found that *IGFBP5* was highly expressed in the LL stage (Appendix A), which may be closely related to the extensive apoptosis of mammary cells at late lactation. In addition, the study found that the expression of *IGFBP5* was up-regulated during the first 2 weeks of involution in dairy cows [92]. The expression pattern of *IGFBP5* in the goat mammary gland in this study is similar to that described in the above study [92]. Furthermore, MMP activity remains low during lactation, as the secretory phenotype requires an intact basement membrane; however, MMPs are highly expressed after weaning [93]. *MMP2* can promote terminal bud invasion by inhibiting epithelial cell apoptosis at the onset of puberty. *MMP2* is also enhanced during bovine mammary gland involution, has properties to degrade basement membrane collagen, and has gelatinolytic activity [94]. *MMP19* is strongly expressed in the myoepithelial layer of the ductal system, and *MMP19* plays a role in mammary angiogenesis [95]. The present study also found a high expression of *MMP2* and *MMP19* in the dry period and involution period. In addition, *VEGFA* encodes a heparin-binding protein, a growth factor that induces the proliferation and migration of vascular endothelial cells. *PDGFRB* overexpression upregulates *VEGFA* and may play a role in mammary angiogenesis by promoting VEGF signaling to induce endothelial cell proliferation. These genes may be a useful target for enhancing blood flow, thereby providing nutrients to the mammary glands for milk production. *PDGFRB* overexpression also affected the expression of *JAK2*, *STAT1*, *IRS2*, etc. However, *PDGFRB* overexpression had no effect on genes encoding milk proteins (*CSN1S2*, *LALBA*, *CSN3*), *PRLR*, and *MAPK1*. Taken together, these suggest that overexpression of *PDGFRB* during involution may affect the expression of genes related to apoptosis and tissue remodeling in mammary involution.

There are some limitations to this study. First, this study used mammary gland tissue samples from different animal sources. Three breeds of dairy goats (Laoshan, Xinong Saanen, and Murciano-Granadina) were used in this study, which could be a source of variability. Differences in genetic information between animals of different breeds or between individual animals may interfere with the results of differential expression analysis in the mammary gland at different development stages [96,97]. In addition, batch effects can also affect the accuracy of transcriptome sequencing results [98]. Although this study corrected for batch effects of samples using RUVSeq software. Furthermore, a balanced experimental design and appropriate sample size may be helpful in interpreting the correct experimental results. Although a total of 42 breast samples were used in this study, only three replicates were used in the IN and LG periods. A balanced experimental design and replicates using appropriate mammary tissue samples at each developmental stage may lead to more efficient and accurate transcriptome profiling results. This study also filtered some low-expressed transcripts, which may ignore some important information regulating gene expression. Because the expression levels of some transcription factors and non-coding RNAs are low [99,100].

## 4. Materials and Methods

### 4.1. Ethics Statement

This study used four parts of experimental data, of which the first and second were carried out under the supervision and guidance of the Animal Care and Use Committee of Shandong Agricultural University, and the authorization numbers are SDAUA-2018–048 and No. 2004005 respectively. The third was approved by the Ethics Committee on Animal and Human Experimentation of the Universitat Autònoma de Barcelona (procedure code: UAB 3859). The fourth animal study was reviewed and approved by Animal Care and Use Committee of Northwest A&F University.

### 4.2. Collection of Goat Mammary Gland Samples

A total of 42 samples from four parts of the transcriptome datasets of goat mammary gland tissues were used in this study (Figure 1, Appendix A). The datasets cover six periods of mammary gland development: early lactation, peak lactation, late lactation, dry period, involution, and late gestation. Among them, the first dataset is from nine non-lactation mammary gland tissue samples in the previous study from our laboratory [8]. These nine healthy Laoshan dairy goats were raised in Qingdao Aote Farm (Qingdao, Shandong, China) under the same feeding and management environment. The goats ranged in age from 3 years 8 months to 4 years 1 month in third parity. Mammary gland tissue was collected after goats were slaughtered in late lactation (LL; n = 3, 240 days postpartum), dry period (DP; n = 3, 300 days postpartum), and late gestation (LG; n = 3, 140 days post-mating). Total RNA was extracted using TRIzol’s RNAiso Plus kit (Takara, Beijing, China). RNA quality was analyzed using NanoDrop 2000 C (Thermo Fisher Scientific, Wilmington, NC, USA). RNA samples were used for subsequent experimental analysis only when the RNA integrity number was greater than 8. Nine sequencing libraries were constructed using the TruSeq RNA Library Prep Kit v2 (Illumina, San Diego, CA, USA) and sequenced on the HiSeq 2500 platform (Illumina). Sequencing data was submitted to NCBI Gene Expression Omnibus (GEO accession: GSE185981). The second part is also the transcriptome sequencing data of the goat mammary gland samples from our previous study [9], namely five healthy Laoshan dairy goats (four years old, the third lactation) from the Qingdao Aote Farm. Early, peak, and late lactation mammary gland tissues were surgically collected by biopsies, and transcriptome sequencing was performed by Illumina HiSeq 2000 (GEO accession: GSE135930). The third part of the data is from the mammary gland tissue of seven Murciano-Granadina goats [35]. The average age of the sampled goats was 5.88 ± 1.89 years and none of them was pregnant. Goat mammary gland tissue samples were collected by biopsies at 78.25 ± 9.29 days (T1, early lactation), 216.25 ± 9.29 days (T2, late lactation), and 285.25 ± 9.29 days (T3, dry period) and sequenced on the Illumina HiSeq 4000 platform (SRA Accession: PRJNA607923). The fourth part of the data is from the mammary gland tissue of nine healthy Xinong Saanen dairy goats (approximately 3 to 4 years old from 2 to 3 parities) [10]. Goat mammary gland tissues during lactation (100 days postpartum, peak lactation), cessation of milking (310 days postpartum, dry period), and involution phase (non-lactation and non-pregnant period) were collected after slaughter and transcriptome sequenced by Illumina HiSeq 2000 (SRA Accession: PRJNA637690).

### 4.3. Sequencing Data Analysis

All samples were quality checked using FastQC software (version 0.11.9, Babraham Bioinformatics, Cambridge, UK) [101], and high-throughput sequencing reads were processed using Trimmomatic software (version 0.39, USADEL Lab, Jülich, Germany) to remove adapter sequences, primers, poly-A tails, and low-quality sequences [102]. Construction of the index of the goat reference genome (BioSample: SAMN03863711) and alignment of sequencing data was done via HISAT2 software (version 2.2.1, Kim Lab, Dallas, TX, USA) [103]. The sam files aligned with the reference genome were converted to bam format files and sorted using samtools software (version 1.10, Wellcome Trust Sanger Institute, Cambridge, UK) [104]. Transcript splicing and quantification were accomplished using StringTie software (version 2.2.0, Johns Hopkins University, Baltimore, MD, USA) [105]. Use the R package RUVSeq (version 1.32.0, University of California, Berkeley, Berkeley, CA, USA) to filter the raw data based on expression levels, by requiring more than 5 reads in at least two samples for each gene. The samples were corrected using the RUVs function in RUVSeq; estimating the factors of unwanted variation using replicate samples, followed by principal component analysis and relative logarithm expression analysis between samples was done using this software [106].

### 4.4. Weighted Gene Co-Expression Network Analysis

To further screen for genes closely associated with each developmental stage, we performed a weighted co-expression analysis of transcripts using the R package WGCNA (version 1.71, University of California, Los Angeles, CA, USA) [41]. Missing values in the data were first detected using the goodSamplesGenes function. Subsequently, the Pearson correlation matrix for all possible RNA pairs was calculated, which was then converted into an adjacency matrix with a soft thresholding function using the “picksoftThreshold” function and filtered for soft thresholding. The co-expression matrix was constructed using a one-step method via the blockwiseModules function, where minModuleSize = 30 and the merged module threshold was set to 0.7. The hierarchical clustering tree corresponding to all transcripts and the corresponding modules were drawn using the plot Dendro And Colors. The module eigengene function was used to compute the module eigengenes (first principal components) of a module in a given single dataset. Correlation and significance test values of module eigengenes and each developmental time period were calculated using the function cor and corPvalueStudent, respectively. When *p* < 0.05 was set for statistical significance. A heatmap of the relationship between modules and traits was presented using the labeledHeatmap function. At the same time, we calculated gene significance (GS), and used plotModuleSignificance to draw box plots for display. The eigengene-based connectivity was calculated by the signedKME function, also known as module membership (MM). The relationship between GS and MM in each module was shown by the verboseScatterplot function. A heatmap of the relationship between modules and traits was presented using the labeledHeatmap function. At the same time, we calculated gene significance (GS) and used plotModuleSignificance to draw box plots for display. The eigengene-based connectivity is calculated by the signedKME function, also known as module membership (MM). The relationship between GS and MM in each module was shown by the verboseScatterplot function.

### 4.5. Differential Expression Analysis and Gene Expression Pattern Analysis

Differential expression analysis was performed using the R package limma (version 3.54.0, The University of Melbourne, Parkville, Australia) [107], and transcripts were considered differentially expressed when *p*.adj (adjusted *p*-value) < 0.05 and log2 fold change ≥ 1. Heatmaps were drawn using the R package pheatmap to show the expression patterns of all differentially expressed transcripts. The number of differentially expressed transcripts between different comparison groups was counted by the R package UpSetR (version 1.4.0, Harvard Medical School, Boston, MA, USA).

### 4.6. Gene Ontology (GO) and Kyoto Encyclopedia of Genes and Genomes (KEGG) Pathway Enrichment Analysis

All differentially expressed genes were subjected to GO functional annotation and KEGG pathway enrichment analysis using clusterProfiler software (version 4.6.0, Southern Medical University, Guangzhou, China), and the term or pathway was considered significantly enriched when *p*.adj < 0.05 [108]. Genes of goat annotated in GO and KEGG databases were used as background genes for enrichment analysis. Based on the results of gene function analysis, genes related to mammary gland development, apoptosis and autophagy, immunity, and material metabolism were screened. The protein-protein interaction (PPI) network of these genes was analyzed by STRING software (version 11.5, European Molecular Biology Laboratory, Heidelberg, Germany). The degree of each gene was calculated and the PPI network was drawn by Cytoscape software (version 3.9.1, Institute for Systems Biology, Seattle, WA, USA).

### 4.7. Goat Mammary Epithelial Cells Culture

Goat mammary gland tissue was obtained surgically. After rinsing the mammary tissue with sterile Phosphate-buffered saline (PBS), fat and connective tissue was removed with ophthalmic scissors, and cut into pieces (0.5–1 mm^3^). The primary goat mammary epithelial cells were then cultured using the tissue block method in saturated humidity at 37 °C with a gas concentration of 5% CO_2_ [109]. The medium components used for culturing goat mammary epithelial cells include: in Dulbecco’s Modified Eagle Medium/Nutrient Mixture F-12 (DMEM/F-12) (Gibco, Thermo Fisher Scientific, Wilmington, NC, USA) supplemented with 10% fetal bovine serum (Gibco), 0.25 mM hydrocortisone, 5 mg/mL insulin (Sigma-Aldrich, St. Louis, MO, USA), 50 U/mL penicillin/streptomycin (Sigma-Aldrich), and 10 ng/mL epidermal growth factor 1 (PrimeGene, Shanghai, China). Subsequently, mammary epithelial cells were seeded in 6-well culture plates at a seeding density of 3 × 10^5^. When cells reach approximately 75% confluence, prepare for cell transfection.

### 4.8. Cell Transfection

A small interfering RNA (siRNA) targeting *PDGFRB* was designed based on the coding sequence of goat *PDGFRB* (accession number: XM_018050163.1) and was commissioned to be synthesized by RiboBio (Guangzhou, China). The sequences of siRNA-PDGFRB were as follows: sense/passenger strand (5’-3’): CGAAAACGGCUCUACAUCUUU and guide/antisense strand (5’-3’): AGAUGUAGAGCCGUUUUCGCU. The PDGFRB cDNA sequence was cloned into pcDNA3.1 vector (Sigma-Aldrich) to increase PDGFRB mRNA and protein expression levels. Empty siRNA vector (si-NC), *PDGFRB*-interfering vector (si-*PDGFRB*), empty *PDGFRB* vector (NC), and *PDGFRB* overexpression vector were transfected into goat mammary epithelial cells cultured in vitro respectively. The specific transfection procedure was performed using Lipofectamine 3000 kit (Thermo Fisher Scientific, Wilmington, NC, USA). Cell transfection was performed when goat mammary epithelial cells had grown to 75% confluence. First, dilute Lipofectamine 3000 Reagent with Opti-MEM™ medium and mix well. Next, prepare a DNA master mix by diluting DNA in Opti-MEM™ Medium, then add P3000™ Reagent and mix well. Note, do not add P3000™ Reagent when diluting siRNA. In the third step, add diluted DNA (1:1 ratio) to each tube of diluted Lipofectamine™ 3000 reagent. Incubate at room temperature for 15 min. Finally, add DNA-lipid complex to goat mammary epithelial cells. Cells were incubated for 48 h at 37 °C, and then transfected cells were analyzed. Expression levels of *PDGFRB* were analyzed by western blotting 48 h after cell transfection.

### 4.9. Cell Proliferation Assay

Goat mammary epithelial cells were seeded in 6-well plates at a density of 3 × 10^5^ cells per well. When the cells reached 75% confluence, the cells were transfected with the overexpression vector (*PDGFRB*) and the interference vector (si-*PDGFRB*), respectively. The specific transfection procedure was performed using the standard protocol supported by the Lipofectamine 3000 kit (Thermo Fisher Scientific). 48 h after the cells were transfected, 5-Ethynyl-2′-deoxyuridine (EdU) staining solution (Beyotime, Beijing, China) was added to each well at a final concentration of 10 μM per well, and then the cells were incubated at 37 °C for 2 h. After the EdU-labeled cells were completed, the culture medium was removed and washed three times with PBS for 5 min each. Add 4% paraformaldehyde to fix for 20 min, and then wash with PBS three times for 5 min each time. Remove the wash solution and incubate at room temperature for 10–15 min with 1 mL of permeabilization solution per well. Remove the permeabilizer and wash the cells 1–2 times with 1 mL of washing solution per well, 3–5 min each time. Add 0.5ml of Click reaction solution to each well, shake the plate gently to ensure that the reaction mixture can evenly cover the sample, and incubate at room temperature for 30 min in the dark. Aspirate the Click reaction solution and wash with washing solution 3 times for 3–5 min each time. Add 1 mL of 1X Hoechst 33342 solution (Beyotime) to each well and incubate at room temperature for 10 min in the dark. Aspirate the 1X Hoechst 33342 solution and wash it 3 times with the washing solution for 3–5 min each. Mount the slides with an anti-fluorescence quenching mounting medium, observe, and take pictures under a microscope (Olympus Model IX51).

### 4.10. Apoptosis and Cell Cycle Detection Assays

Goat mammary epithelial cells were cultured using 6-well plates. Forty-eight hours after cell transfection, 5 µL of annexin V-fluorescein isothiocyanate (FITC) and 10 µL of propidium iodide (PI) staining solution (Invitrogen, Thermo Fisher Scientific, Wilmington, USA) were added to each well and incubated for 10 min at room temperature in the dark. Apoptosis detection was performed using a BD flow cytometer (BD Biosciences, Franklin Lakes, USA). The content of DNA in the cells was detected by PI staining and the cell cycle was analyzed. FlowJo software (version 7.6.1, BD Biosciences) was used to analyze the flow cytometry results. The protein expression levels of CCND1, caspase-3, Bax and Bcl-2 were detected by Western blotting 48 h after transfection.

### 4.11. Western Blotting

Radioimmunoprecipitation assay buffer (RIPA buffer) (Beyotime) was used to extract total protein from mammary epithelial cells. The protein concentration of the extracts was measured at 562 nm using a bicinchoninic acid (BCA) assay kit (Beyotime) on a microplate reader (Molecular Devices, San Jose, CA, USA). Prepare 10% separating gel and 5% stacking gel (Beyotime). 20 μg of total protein from each sample was separated by SDS-PAGE and the separated protein was transferred to the polyvinylidene fluoride (PVDF) membrane. Each PVDF membrane was blocked for 1.5 h in a buffer containing 5% nonfat dry milk, then incubated in a buffer containing primary antibodies against the target protein for 12 h at 4 °C. After 3 washes with 1x Tris-buffered saline containing Tween 20, the PVDF membrane was incubated with the appropriate secondary antibody for 50 min at room temperature. Membranes were then infiltrated with ECL chemiluminescence buffer (Beyotime) for 1 min and immunoreactive proteins were detected using an Azure 300 Chemiluminescence Western Blot Imaging System (Azure Biosystems, Dublin, CA, USA). Protein expression levels were calculated using ImageJ 1.48 software (National Institutes of Health) with GAPDH as an internal reference for expression level correction. The antibodies used in this study included anti-PDGFRB (1:1000 dilution, ab69506, Abcam, Shanghai, China), anti-p53 (1:1000 dilution, ab26, Abcam), anti-p21(1:1000 dilution, ab109199, Abcam), anti-CCND1 (1:1500 dilution, #55506, CST), anti-GAPDH (1:2000 dilution, #2118, CST), anti-PI3K (1:1000 dilution, #4257, CST), anti-p-PI3K (1:1000 dilution, #17366, CST), anti-Akt (1:1000 dilution, #4691, CST), anti-p-Akt (1:1000 dilution, #4060, CST), anti-caspase-3 (1:2000 dilution, #14220, CST), anti-BCL-2 (1:1000 dilution, ab182858, Abcam), anti-Bax (1:1000 dilution, #2772, CST) and HRP-conjugated goat antirabbit IgG (1:2000, CW0103, CWBIO, Beijing, China).

### 4.12. Real-Time Quantitative Reverse-Transcription PCR Assay

Gene primer sequences were designed using Primer-BLAST of the National Center for Biotechnology Information [110]. RNA was extracted using the TRIzol method. Reverse transcription and PCR amplification were performed using the One Step TB Green PrimeScript RT-PCR Kit (Takara). RT-qPCR was performed on LightCycler 96 (Roche, Basel, Switzerland), a real-time PCR system. Use NormFinder software (version 5, Aarhus University Hospital, Skejby, Denmark) to select the most stable gene pairs from the 6 genes as reference genes for RTqPCR assay [111]. The primer amplification efficiency was calculated by establishing a standard curve. Use the geometric mean of two carefully selected reference genes (*GAPDH* and *MRPL39*) as an accurate normalization factor [112]. Relative gene expression calculated by the modified Pfaffl method [8,113]. The sequences and amplification efficiencies of the gene primers are provided in Appendix A.

### 4.13. Statistical Analysis

Student’s *t*-test was used to test the significance of the difference between the experimental group and the control group; *p* < 0.05 indicated a significant difference. Tukey’s honest significant difference test was used to analyze gene expression levels at different developmental stages. Figures in this work were generated using R software (version 4.2.0; https://www.r-project.org/; accessed on 5 May 2022) unless otherwise stated.

## 5. Conclusions

This study analyzed the mammary transcriptome profile of goats during lactation and non-lactation. Twelve gene co-expression networks associated with expression profiles in the mammary glands of dairy goats at six different developmental stages were constructed. Analysis of gene function in the network found that immune-related genes in the mammary gland were more actively expressed in the early, dry, and involutional stages, thereby strengthening the protection of the health of non-lactation mammary glands. The expression of genes related to substance metabolism and transport is active in the late pregnancy and peak lactation period, which ensures the secretion of milk by the mammary glands. Differential expression analysis revealed that genes at the transcriptional level adapt to mammary gland structure and function at different developmental stages. This is manifested in that gene expression patterns related to cell growth, apoptosis, immunity, substance transport, biosynthesis, and metabolism are significantly altered during mammary gland development and involution. Members of the solute carrier family play an important role in the material transport process in the mammary gland, and the PPAR signaling pathway is an important pathway for regulating lipid metabolism and milk fat production. Notably, *PDGFRB* was screened as a hub gene of the mammary gland development network, which can slow down the G1/S phase arrest of the goat mammary epithelial cell cycle and promote mammary epithelial cell proliferation by regulating the PI3K-Akt signaling pathway. In addition, *PDGFRB* overexpression can also affect the expression of apoptosis, matrix metalloproteinase family, vascular development, and other related genes, which is beneficial to the remodeling of mammary gland tissue during involution. In conclusion, the genes and signaling pathways identified in this study will provide a theoretical basis for future research on mammary gland development, function, disease, molecular breeding, and feeding management.

## Figures and Tables

**Figure 1 ijms-23-14424-f001:**
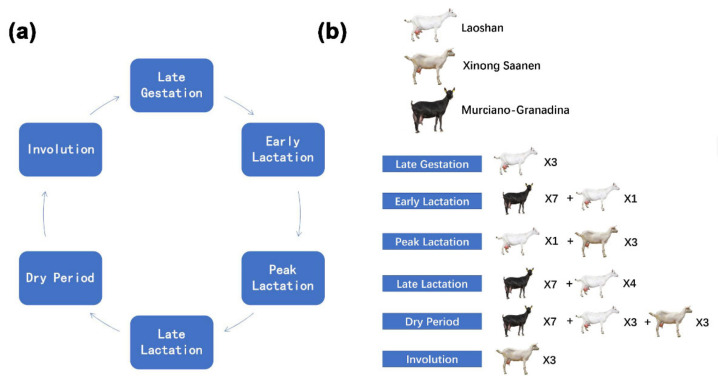
The number and breed distribution of dairy goats during six different mammary developmental stages. (**a**) Pattern diagram of the six mammary gland developmental stages in this study. (**b**) The number and breed of goats during different mammary developmental stages.

**Figure 2 ijms-23-14424-f002:**
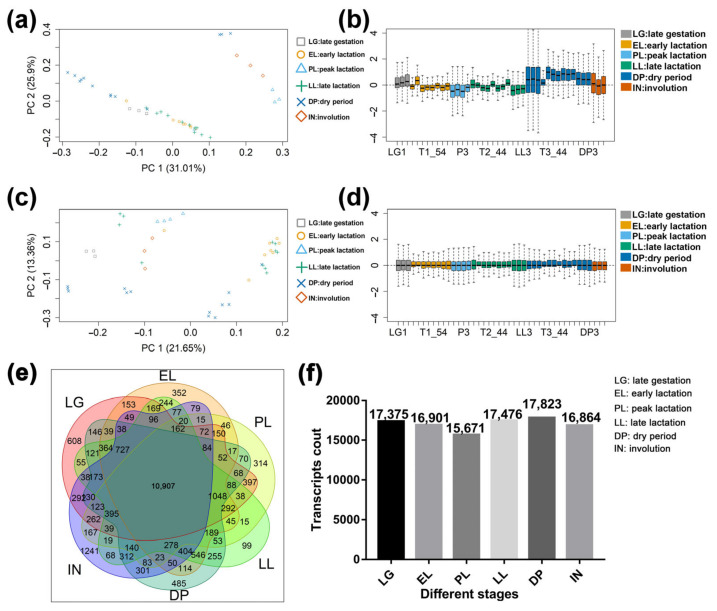
Quality assessment and basic statistics of the sequencing data. (**a**) Principal component analysis of raw mammary gland samples. (**b**) The boxplots of relative log expression (RLE = log-ratio of read count to median read count across sample) of raw mammary gland samples. (**c**) Principal component analysis of processed mammary gland samples. (**d**) The boxplots of relative log expression (RLE = log-ratio of read count to median read count across sample) of processed mammary gland samples. (**e**) Venn plot of expressed transcripts (FPKM ≥ 2) in mammary gland tissue at different developmental stages. (**f**) Number of expressed transcripts in the mammary gland at each developmental stage. EL: early lactation. PL: peak lactation. LL: late lactation. DP: dry period. IN: involution stage. LG: late gestation.

**Figure 3 ijms-23-14424-f003:**
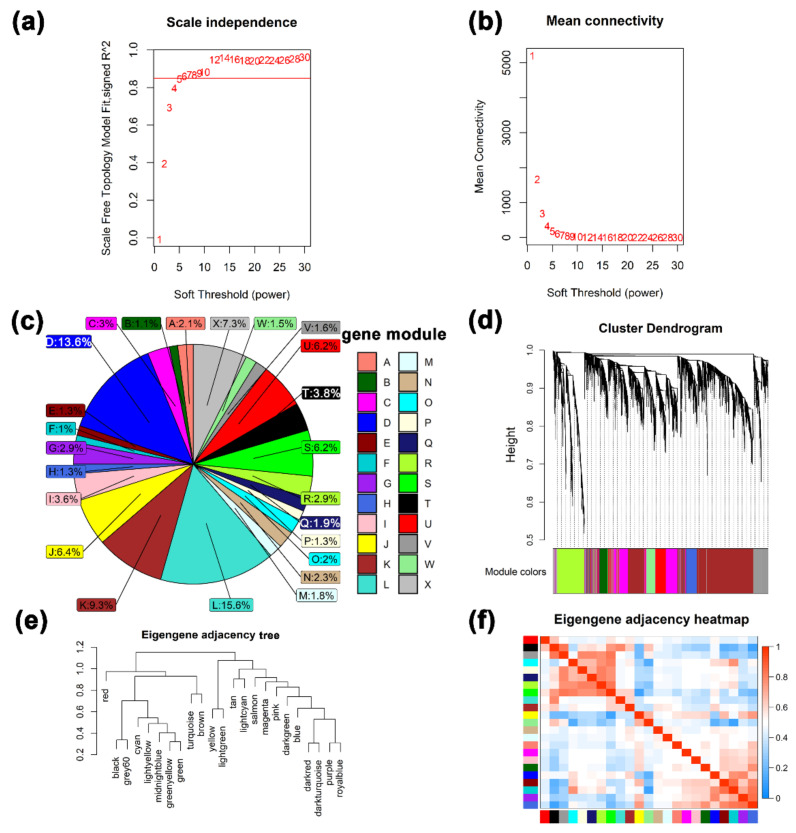
The results of weighted gene correlation network analysis (WGCNA). (**a**) and (**b**) Soft-thresholding power analysis was used to obtain the scale-free fit index of network topology. (**c**) A pie chart of the distribution of the number of genes in each color module. (**d**) Hierarchical cluster analysis was conducted to detect co-expression clusters with corresponding color assignments. Each color represents a module in the constructed gene co-expression network by WGCNA. (**e**) and (**f**) the relationship among different modules.

**Figure 4 ijms-23-14424-f004:**
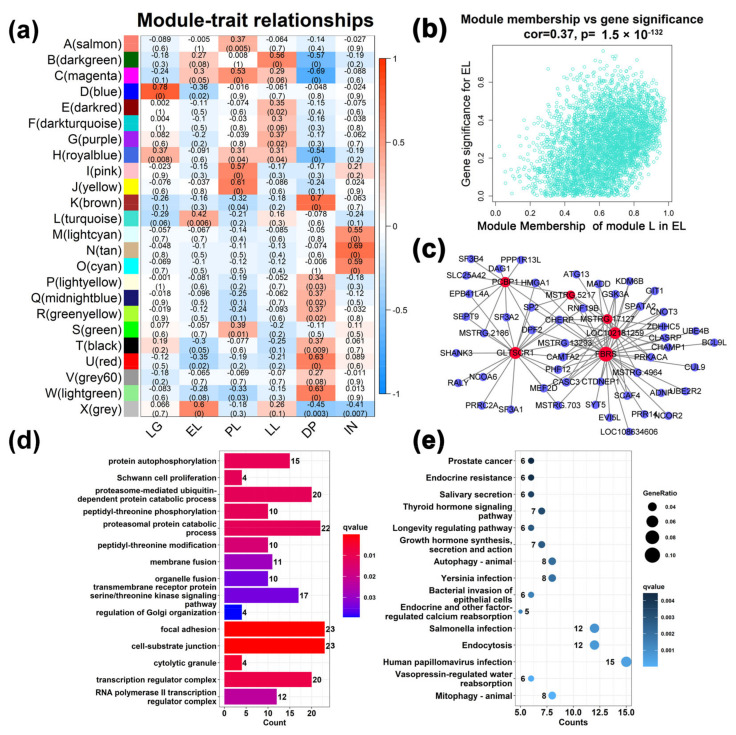
Module-different stages associations. (**a**) Each row corresponds to a module eigengene, and each column corresponds to a mammary gland development stage. Each cell contains the corresponding correlation in the first line and the *p*-value in the second line. The table is color-coded by correlation according to the color legend. (**b**) A scatterplot of Gene Significance (GS) for the early lactation (EL) stage vs. Module Membership (MM) in the L (turquoise) module. There is a significant correlation (cor = 0.37, *p* < 0.05) between GS and MM in this module. (**c**) Visualization of connections of genes in L (turquoise) module. According to the connection weight value, only the first 100 functional relationships are shown in the figure. (**d**) Gene Ontology (GO) enrichment analysis of genes in the module L associated with the EL. (**e**) Kyoto Encyclopedia of Genes and Genomes (KEGG) pathway enrichment analysis of genes in the module L associated with the EL.

**Figure 5 ijms-23-14424-f005:**
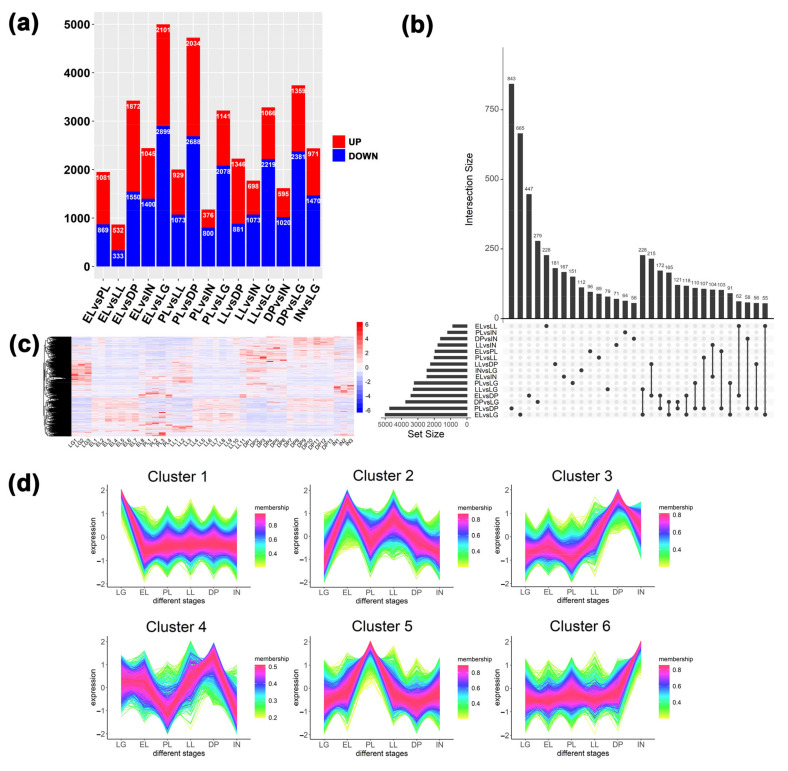
Differential expression analysis and expression pattern analysis of transcripts in mammary glands at different stages. (**a**) Statistical results of the number of up- and down-regulated transcripts in different comparison groups. (**b**) Statistical analysis of the distribution of differentially expressed transcripts between different comparison groups. (**c**) Expression heatmap of all differentially expressed transcripts. (**d**) Analysis of expression patterns of all differentially expressed transcripts.

**Figure 6 ijms-23-14424-f006:**
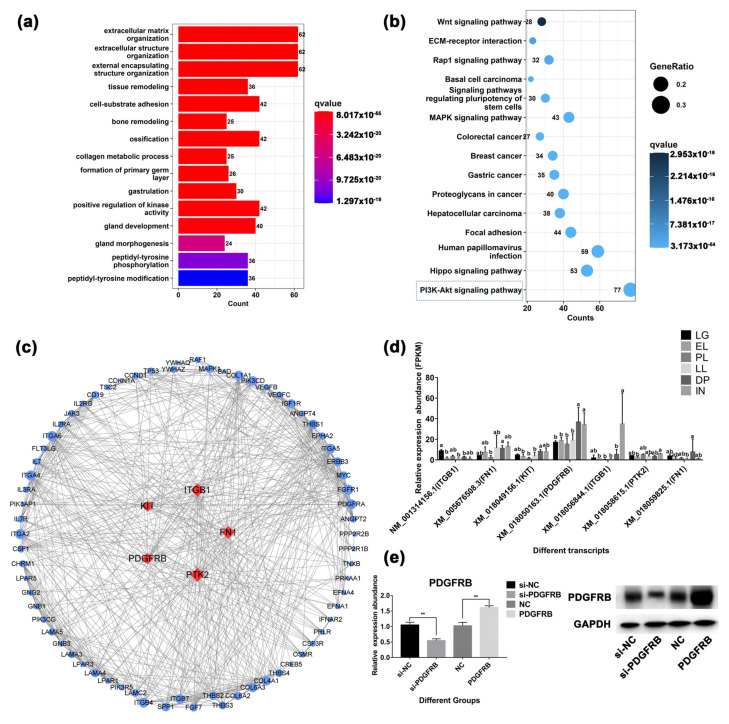
Mammary gland development-related gene analysis. (**a**) Gene Ontology (GO) analysis of 226 genes associated with mammary gland development. (**b**) Kyoto Encyclopedia of Genes and Genomes (KEGG) pathway enrichment analysis of genes associated with mammary gland development. (**c**) Protein-protein interaction analysis of genes enriched in PI3K-Akt signaling pathway. (**d**) Analysis of expression patterns of core genes at different developmental stages. Lowercase letters represent significance at the 0.05 level. (**e**) The results of interference or overexpression of *PDGFRB* in mammary epithelial cells cultured in vitro by Western blotting. si-NC: native control of small interfering RNA (empty vector of siRNA); si-*PDGFRB*: the small interfering RNA targeting *PDGFRB*; NC: native control (empty vector of pcDNA3.1); *PDGFRB*: *PDGFRB* overexpression vector. ** indicates *p* < 0.01, NS indicates not significant, the data are presented as the mean ± SEM (n = 3).

**Figure 7 ijms-23-14424-f007:**
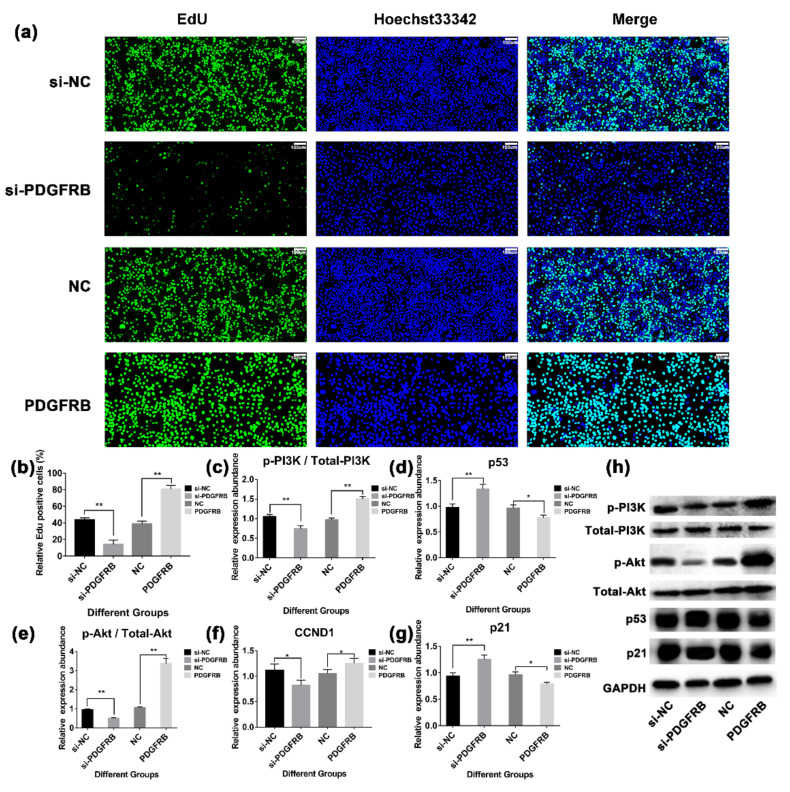
Effect of overexpression or interference with the expression of *PDGFRB* in mammary epithelial cells on the PI3K-Akt signaling pathway. (**a**) The proliferation of mammary epithelial cells was detected by EdU (5-ethynyl-2′-deoxyuridine) staining assay. (**b**) The bar plot shows statistics of cell proliferation for different groups. (**c**–**g**) The bar plots show the expression of PI3K-Akt signaling pathway members after interference or overexpression of *PDGFRB*, respectively. (**h**) Western blotting was used to detect the expression of PI3K-Akt signaling pathway members after interference or overexpression of *PDGFRB*. si-NC: native control of small interfering RNA (empty vector of siRNA); si-*PDGFRB*: small interfering RNA targeting *PDGFRB*; NC: native control (empty vector of pcDNA3.1); *PDGFRB*: *PDGFRB* overexpression vector. * indicates *p* < 0.05, ** indicates *p* < 0.01, NS indicates not significant, the data are presented as the mean ± SEM (n = 3).

**Figure 8 ijms-23-14424-f008:**
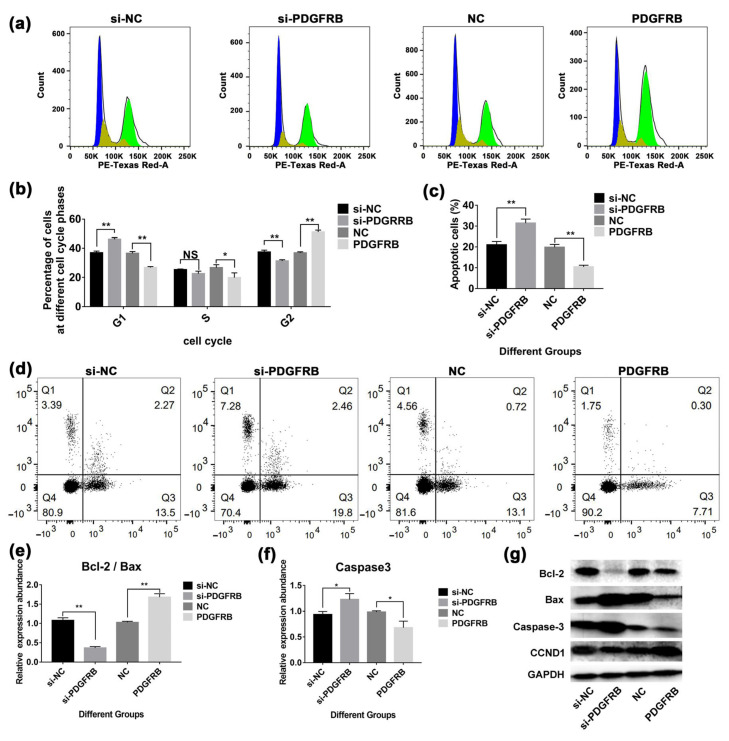
Effect of overexpression or interference with the expression of *PDGFRB* in mammary epithelial cells on apoptosis and the cell cycle. (**a**) Analysis of cell cycle assays. Forty-eight hours after cell transfection, the cells were stained with PI and incubated in the dark for 10 min for cell cycle detection using BD flow cytometry. (**b**) The bar graph shows the proportion of cells at different stages of the cell cycle. (**c**) and (**d**) Analysis of apoptotic goat mammary epithelial cells. Flow cytometry was used to detect the apoptosis of mammary epithelial cells after overexpression and interference with PDGFRB. Forty-eight hours after the cells were transfected, the cells were stained with annexin V-FITC/propidium iodide (PI), incubated in the dark for 10 min, and then detected using a BD flow cytometer. The bar graph shows the statistics of apoptotic cells in the different groups (si-*PDGFRB* vs. si-NC, *PDGFRB* vs. NC). (**e**–**g**) Analysis of apoptosis-related protein expression. The expression of proapoptotic (caspase-3, Bax), antiapoptotic (Bcl-2), and cyclin (CCND1) genes was detected by Western blotting. The bar graphs show the ratio of Bcl-2/Bax and the expression of Caspase-3, respectively. (si-NC: native control of small interfering RNA (empty vector of siRNA); si-*PDGFRB*: the small interfering RNA targeting *PDGFRB*; NC: native control (empty vector of pcDNA3.1); *PDGFRB*: *PDGFRB* overexpression vector. * indicates *p* < 0.05, ** indicates *p* < 0.01, NS indicates not significant, the data are presented as the mean ± SEM (n = 3).

**Figure 9 ijms-23-14424-f009:**
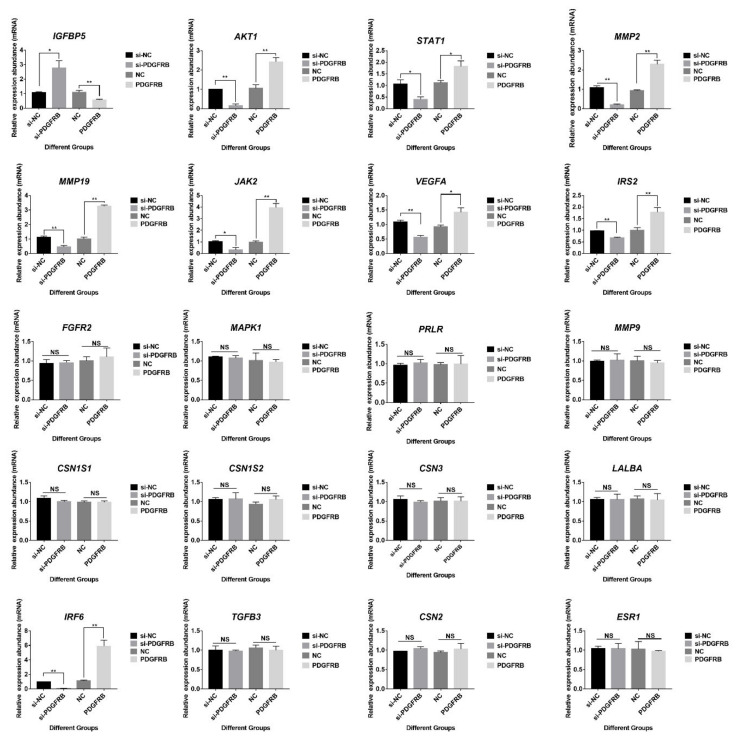
Effect of *PDGFRB* on genes related to mammary gland development and involution. Reverse transcription quantitative PCR was used to detect the expression of mammary gland involution-related genes in mammary epithelial cells after overexpression or interference with *PDGFRB*. The bar graphs show the relative expression of 20 genes at transcription level in different groups (si-*PDGFRB* vs. si-NC; *PDGFRB* vs. NC). si-NC: native control of small interfering RNA (empty vector of siRNA); si-*PDGFRB*: the small interfering RNA targeting *PDGFRB*; NC: native control (empty vector of pcDNA3.1); *PDGFRB*: *PDGFRB* overexpression vector. * indicates *p* < 0.05, ** indicates *p* < 0.01, NS indicates not significant, data are presented as mean ± SEM (n = 3).

## Data Availability

The data presented in this study are available on request.

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
