# Peer review of "Transcriptome Analysis of Goat Mammary Gland Tissue Reveals the Adaptive Strategies and Molecular Mechanisms of Lactation and Involution"

_ijms, 2022, doi:10.3390/ijms232214424_

Round 1

Reviewer 1 Report

The manuscript presents the transcriptome sequencing results of goat mammary gland tissue at different developmental stages. The authors also investigated genes related to cell growth, apoptosis, immunity, nutrient transport, and synthesis that might be involved in the metabolic regulation of mammary lactation. In addition, they overexpressed the PDGFRB gene and suggest that PDGFRB is important for gland tissue involution by affecting a series of genes expression related to apoptosis, matrix metalloproteinase family, vascular development, etc.

The work appears to have been done carefully. The manuscript is well written and the results are correctly presented. The outcome might be useful for people working in mammary gland development.

Apart from that, I have some minor questions and concerns,

It is better to describe the developmental stage chronologically. Late gestation should be before late lactation. Also, it is good to describe the stages in the Introduction session.

The text resolution of all figures is too low to read, especially for the numbers.

Line 283-287, the introduction of PDGFRB should be moved to the Introduction session.

Line 634, 9+5+7+9=30 samples, where are 42 samples coming from? Please indicate it.

Line 743, how did the vectors transfect the epithelial cells? Is it by only the vectors with the standard protocol or by some virus for the infection? Please describe it more clearly.

Author Response

The manuscript presents the transcriptome sequencing results of goat mammary gland tissue at different developmental stages. The authors also investigated genes related to cell growth, apoptosis, immunity, nutrient transport, and synthesis that might be involved in the metabolic regulation of mammary lactation. In addition, they overexpressed the PDGFRB gene and suggest that PDGFRB is important for gland tissue involution by affecting a series of genes expression related to apoptosis, matrix metalloproteinase family, vascular development, etc. The work appears to have been done carefully. The manuscript is well written and the results are correctly presented. The outcome might be useful for people working in mammary gland development.

Response 0: Thank you for your comments. We thank the reviewers for their acknowledgment of this study.

1.It is better to describe the developmental stage chronologically. Late gestation should be before late lactation. Also, it is good to describe the stages in the Introduction session.

Response 1: Thank you for your comments. We have rewritten the Introduction section to follow the order from late gestation to involution (lines 32-57, lines 87-88). At the same time, we also modified the order in the resulting images (Figure 1, Figure 2e and 2f, Figure 5c and 5d, Figure 6d).

2.The text resolution of all figures is too low to read, especially for the numbers.

Response 2 Thank you for your comments. We have revised all images in the manuscript for clearer reading.

3.Line 283-287, the introduction of PDGFRB should be moved to the Introduction session.

Response 3: Thank you for your comments. We have removed the introduction of PDGFRB to the Introduction session in the revised manuscript (lines 91-94).

4.Line 634, 9+5+7+9=30 samples, where are 42 samples coming from? Please indicate it.

Response 4: Thank you for your comments. This study did use 42 mammary gland samples. We describe the distribution of goat breeds and numbers at different stages of mammary gland development in Figure 1. In addition, Table S2 provides source and grouping information for all mammary gland samples.

5.Line 743, how did the vectors transfect the epithelial cells? Is it by only the vectors with the standard protocol or by some virus for the infection? Please describe it more clearly.

Response 5: Thank you for your comments. Vector transfection followed the standard protocol of the Lipofectamine 3000 kit (Thermo Fisher Scientific). We have supplemented the specific transfection steps in Sections 4.8 (lines 882-888) and 4.9 (lines 894-895) in Materials and methods.

Reviewer 2 Report

In the manuscript entitled “Transcriptome analysis of goat mammary gland tissue reveals the adaptive strategies and molecular mechanisms of mammary gland development and involution”, the authors investigated the evolution of the mammary gland mRNA profiles during different physiological stages from late gestation to involution.  

The study is interesting and provides an overview of the mammary gland transcriptomes. However, the manuscript needs some modifications.  

1.       As the authors described the development of the mammary gland occurs in different growth cycles starting very early. They studied several periods during lactation and only one period at the end of gestation. Thus, the title indicating a study on mammary gland development should be changed to be more appropriate to the reported study.  Similarly, the introduction is very interesting but does not fit with the reported work. This section should be revised.

2.       The authors present the different stages from the start of lactation to the end of gestation. It would be more relevant to start with the end of gestation until the involution even if lactation cycles are a cyclical phenomenon.

3.       The section 2.2 is not easy to follow with the name of the modules corresponding to a color. It would be easier with a name (as A, B, …) in the figure and the text.   

4.       In the section 2.3, the authors must indicate the % of the modules they discussed to allow an evaluation of their importance.

5.       Line 536: revised the sentence. PPAR regulates the genes mentioned.

6.       The discussion must be restructured with sub-paragraphs

7.       Check the references (ex: ref 88-90, 91 (lines 617 & 618) are not appropriate, …)

8.       In the discussion about the limitations of the study, the authors must add the breed of the goats, which could be a source of variability. They must also add the LL and DP period represented by only three biological replicates (line 621)

9.       A figure or table showing all samples, their sources and all the relevant details should be added to clarify the experimental design

10.   Are the mammary gland samples used obtained by biopsies or after slaughter?

11.   Does SDAUA-2018-048 correspond to the agreement number obtained for this experimentation procedures. If this is not the case, the authors must add the number corresponding to the reported animal experiment.  

12.   Regarding the apoptosis observed during involution, a larger discussion on different program cell death (apoptosis and autophagic) should be added: Majeski & Dice Int J Biochem Cell Biol. 2004/ Bursch et al J Cell Sci. 2000/ Ollier et al J Nutr. 2007/ Gajewska et al J Physiol Pharmacol. 2005.

·      

Author Response

In the manuscript entitled “Transcriptome analysis of goat mammary gland tissue reveals the adaptive strategies and molecular mechanisms of mammary gland development and involution”, the authors investigated the evolution of the mammary gland mRNA profiles during different physiological stages from late gestation to involution. The study is interesting and provides an overview of the mammary gland transcriptomes. However, the manuscript needs some modifications.

Response 0: Thank you for your comments. We thank the reviewers for their acknowledgment of this study.

1.As the authors described the development of the mammary gland occurs in different growth cycles starting very early. They studied several periods during lactation and only one period at the end of gestation. Thus, the title indicating a study on mammary gland development should be changed to be more appropriate to the reported study.  Similarly, the introduction is very interesting but does not fit with the reported work. This section should be revised.

Response 1: Thank you for your comments. The title has been revised to “Transcriptome analysis of goat mammary gland tissue reveals the adaptive strategies and molecular mechanisms of lactation and involution”. (lines 1-3). We have rewritten the introduction. (lines 31-123)

2.The authors present the different stages from the start of lactation to the end of gestation. It would be more relevant to start with the end of gestation until the involution even if lactation cycles are a cyclical phenomenon.

Response 2: Thank you for your comments. We have rewritten the Introduction section to follow the order from the start of late gestation to involution (lines 32-57, lines 87-88). At the same time, we also modified the order in the resulting images (Figure 1, Figure 2e and 2f, Figure 5c and 5d, Figure 6d).

3.The section 2.2 is not easy to follow with the name of the modules corresponding to a color. It would be easier with a name (as A, B, …) in the figure and the text.

Response 3: Thank you for your comments. We have used letters (A-X) to denote different gene modules. Please refer to Section 2.2 (lines 165-186), Figure 3c, and Figure 4a for specific modifications.

4.In the section 2.3, the authors must indicate the % of the modules they discussed to allow an evaluation of their importance.

Response 4: Agree, thank you. We discussed 12 gene modules (accounting for 52.17% of all modules) that were significantly positively correlated with 6 stages (cor value ≥ 0.4, pvalue < 0.05). (lines 196-197)

5.Line 536: revised the sentence. PPAR regulates the genes mentioned.

Response 5: Agree, thank you. We have revised the sentence to “PPAR regulates the genes mentioned above enriched the role of PPAR signaling pathway in goat mammary lipid metabolism.”. (lines 661-662)

6.The discussion must be restructured with sub-paragraphs

Response 6: Sorry, we didn't make changes to the Discussion based on this suggestion. On the one hand, due to time constraints; on the other hand, it is not clear which part or position of the Discussion should be restructured with sub-paragraphs. We look forward to your help, thank you very much.

7.Check the references (ex: ref 88-90, 91 (lines 617 & 618) are not appropriate, …)

Response 7: Agree, thank you. We have deleted the original references (ref 88-90, 91) in this section and re-added the appropriate new references (lines 744-745).

8.In the discussion about the limitations of the study, the authors must add the breed of the goats, which could be a source of variability. They must also add the LL and DP period represented by only three biological replicates (line 621)

Response 8: Agree, thank you. R1: We have added the breed of the goats, which could be a source of variability (line 741-742). This sentence has been modified to “Three breeds of dairy goats (Laoshan, Xinong Saanen, and Murciano-Granadina) were used in this study, which could be a source of variability.” R2: Only 3 biological replicates were used in the IN and LG periods. (lines 749-750)

9.A figure or table showing all samples, their sources and all the relevant details should be added to clarify the experimental design

Response 9: Agree, thank you. We describe the distribution of goat breeds and numbers at different stages of mammary gland development in Figure 1. In addition, Table S2 provides source and grouping information for all mammary gland samples.

10.Are the mammary gland samples used obtained by biopsies or after slaughter?

Response 10: Thank you. This paper used 4 parts of experimental data, in which the first part of mammary gland tissue was obtained after slaughter (GEO accession: GSE185981); the second part was obtained by tissue biopsy (GEO accession: GSE135930); the third part of mammary tissue was obtained by biopsy (SRA Accession: PRJNA607923); the fourth part of mammary tissue was obtained by slaughtering goats (SRA Accession: PRJNA637690). We have made revisions to 4.2. Collection of goat mammary gland samples in the manuscript. (lines 776, 788, 792 and 799).

11.Does SDAUA-2018-048 correspond to the agreement number obtained for this experimentation procedures. If this is not the case, the authors must add the number corresponding to the reported animal experiment.

Response 11: Agree, thank you. Not exactly. This study used four parts of experimental data, of which the first and second were carried out under the supervision and guidance of the Animal Care and Use Committee of Shandong Agri-cultural University, and the authorization numbers are SDAUA-2018–048 and No. 2004005 re-spectively. The third was approved by the Ethics Committee on Animal and Human Experi-mentation of the Universitat Autònoma de Barcelona (pro-cedure code: UAB 3859). The fourth animal study was reviewed and approved by An-imal Care and Use Committee of Northwest A&F University. We have revised 4.1. Ethics Statement in the manuscript (lines 757-767).

12.Regarding the apoptosis observed during involution, a larger discussion on different program cell death (apoptosis and autophagic) should be added: Majeski & Dice Int J Biochem Cell Biol. 2004/ Bursch et al J Cell Sci. 2000/ Ollier et al J Nutr. 2007/ Gajewska et al J Physiol Pharmacol. 2005.

Response 12: Agree, thank you. We supplement the Results (lines 332-345) and Discussion (lines 529-566) on apoptosis and autophagy.

Point 1: I found that sections 2 & 3 should be re‐organized and be shortened. It may be easier for the readers if the authors define properly the mixture of regression model and the class‐ membership equation first before moving to the computation of the GINI and of the Polarization of subgroups. Sections 2.1 and 2.2 are too long and can be significantly reduced. In section 2.1 the authors assume the condition uk > uj, but this does not appear anywhere else in the calculation of the mixture of regression model. After equation (10) all the other equations are not numbered.

Response 1: Please provide your response for Point 1. (in red)

Point 2: The probability for a given country h to be in a class k should be the proportion of observations (households) in country h that belong to the income class k. On page 9, the first equation (it would be easier for the reader if the equation is numbered) is not exactly the proportion of people because the authors take the sum of the probability. The interpretation of the equation in not obvious. Normally, after estimating a mixture of regression model we have for each observation its estimated probabilities to be classified into the different classes identified. What is often done is to classify a given observation into the class where its estimated probability is higher. In many software this is also the method used that gives us the proportion of people in each of the classes. The authors should explain the equation on page 9 and how to interpret it. Alternatively, they may use the proportion approach which will make the interpretation easier.

Response 2: Please provide your response for Point 2. (in red)

Reviewer 3 Report

The study of Xuan et al. profiled the expression characteristics of genes in goat mammary tissue at six stages of lactation and non-lactation, and explored the function of PDGFRB on the PI3K-Akt signaling pathway in goat mammary epithelial cells. This is beneficial for understanding how genes regulate physiological processes such as lactation, mammary gland development, involution, and cellular remodeling. This is a very innovative work, the experimental design is reasonable, and the experimental methods and analysis results are accurate. However, there are still some grammar and format errors, and minor revisions are recommended. The specific amendments are as follows:

1. Abstract: line 17, briefly described the expression patterns of PDGFRB at different developmental stages of mammary glands.

2. The detection of PDGFRB on genes related to mammary gland development and involution was just for mRNA expression, if the WB results were prepared, it will be better.

3. PCA results showed that the six different groups were not divided very well. So, how to explain the differences between the different groups?

4. Line 20-22, this sentence has a grammatical error and should be described as: "In addition, PDGFRB overexpression can also affect the expression of genes related to apoptosis, matrix metalloproteinase family, and vascular development, which is beneficial to the remodeling of mammary gland tissue during involution.”.

5. Line 36, removes spaces before "After partition,".

6. Line 37, add "," before "and the milk ducts transport".

7. Line 46, modified "makes" to "make".

8. Line 49-50, 68. The comma is not in the correct format, please use English format to separate.

9. Line 69-71, please add appropriate references.

10. Line 77-79, revise this sentence to "This will provide theoretical support for treating animal mammary gland diseases and understanding the regulation of mammary gland development and lactation in the future."

11. Line 99, the gene name needs to be written in italics.

12. Line 127, 128, pvalue needs to be written in italics and should be changed to p-value.

13. Line 189, add space in "The heatmap showed".

14. Line 240,243, the gene name should be written in italics.

15. Line 252, not 6 hub genes, should be 5 hub genes.

16. Line 269-270, although the authors provide a table of PPI network information for genes related to mammary gland development in Table S11. However, in order to illustrate this mammary gland development-related network more clearly, please provide the protein interaction network diagram of the 226 genes related to mammary gland development, which can be shown in the attachment.

17. Line 370, 371, gene names should be written in italics.

18. Line 665,669 Please provide the bioinformatics software version used in the manuscript.

Author Response

The study of Xuan et al. profiled the expression characteristics of genes in goat mammary tissue at six stages of lactation and non-lactation, and explored the function of PDGFRB on the PI3K-Akt signaling pathway in goat mammary epithelial cells. This is beneficial for understanding how genes regulate physiological processes such as lactation, mammary gland development, involution, and cellular remodeling. This is a very innovative work, the experimental design is reasonable, and the experimental methods and analysis results are accurate. However, there are still some grammar and format errors, and minor revisions are recommended. The specific amendments are as follows:

1.Abstract: line 17, briefly described the expression patterns of PDGFRB at different developmental stages of mammary glands.

Response 1: Thank you for your comments. We have described the expression pattern of PDGFRB as " Notably, platelet derived growth factor receptor beta (PDGFRB) was screened as a hub gene of the mammary gland developmental network, which is highly expressed during the DP and IN." (lines 19,20)

2.The detection of PDGFRB on genes related to mammary gland development and involution was just for mRNA expression, if the WB results were prepared, it will be better.

Response 2: Thank you for your comments. Your suggestion is very good. As you can see in Figure 1, we don't have mammary tissue samples from the other two breeds of goats (Xinong Saanen and Murciano-Granadina), so unfortunately, western blot assay was not possible.

3.PCA results showed that the six different groups were not divided very well. So, how to explain the differences between the different groups?

Response 3: Thank you for your comments. We agree with you. Indeed, the principal component analysis shows that the distribution of mammary gland samples over different stages is not ideal. The reasons for the poor classification may stem from different breeds of goats and batch effects. We have mentioned the limitations of this study in the Discussion section. In addition, I'm very sorry, to be honest, we don't know how to explain this question, we can only describe the PCA results objectively.

4.Line 20-22, this sentence has a grammatical error and should be described as: "In addition, PDGFRB overexpression can also affect the expression of genes related to apoptosis, matrix metalloproteinase family, and vascular development, which is beneficial to the remodeling of mammary gland tissue during involution.”.

Response 4: Thank you for your comments. We have modified this sentence to “In addition, PDGFRB overexpression can also affect the expression of genes related to apoptosis, matrix metalloproteinase family, and vascular development, which is beneficial to the remodeling of mammary gland tissue during involution”. (lines 22-24)

5.Line 36, removes spaces before "After partition,".

Response 5: Thank you for your comments. We have made extensive revisions to the Introduction section. This sentence has been deleted.

6.Line 37, add "," before "and the milk ducts transport".

Response 6: Thank you for your comments. We have made extensive revisions to the Introduction section. This sentence has been deleted.

7.Line 46, modified "makes" to "make".

Response 7: Thank you for your comments. We have made extensive revisions to the Introduction section. This sentence has been deleted.

8.Line 49-50, 68. The comma is not in the correct format, please use English format to separate.

Response 8: Thank you for your comments. We have modified the comma to the correct English format. Lines 65-66, 78-79.

9.Line 69-71, please add appropriate references.

Response 9: Thank you for your comments. We have supplemented the references in the revised paper. lines 78-79.

10.Line 77-79, revise this sentence to "This will provide theoretical support for treating animal mammary gland diseases and understanding the regulation of mammary gland development and lactation in the future."

Response 10: Thank you for your comments. We have revised this sentence to “This work will provide theoretical support for understanding of the regulation of lactation and mammary involution in the future.”. lines 97-98.

11.Line 99, the gene name needs to be written in italics.

Response 11: Thank you for your comments. We have italicized all gene names in the revised manuscript. Lines 143, 304-306, 310, 320-322, 325, 327, 454, 463, 614, 646, 661.

12.Line 127, 128, pvalue needs to be written in italics and should be changed to p-value.

Response 12: Thank you for your comments. We have written all pvalue in italics in the revised manuscript. Lines 180,181.

13.Line 189, add space in "The heatmap showed".

Response 13: Thank you for your comments. We have added space in “The heatmap showed”. Line 251.

14.Line 240,243, the gene name should be written in italics.

Response 14: Thank you for your comments. We have written all gene names in italics. Lines 143, 304-306, 310, 320-322, 325, 327, 454, 463, 614, 646, 661.

15.Line 252, not 6 hub genes, should be 5 hub genes.

Response 15: Thank you for your comments. We have revised the sentence to “A total of 60 immune-related genes were screened (Figure S18), and five genes were identified as hub genes.” (Line 316)

16.Line 269-270, although the authors provide a table of PPI network information for genes related to mammary gland development in Table S11. However, in order to illustrate this mammary gland development-related network more clearly, please provide the protein interaction network diagram of the 226 genes related to mammary gland development, which can be shown in the attachment.

Response 16: Thank you for your comments. According to your suggestion, we used Cytoscape software to map the protein-protein interaction network of genes related to mammary gland development. The results are supplemented in Figure S25.

17.Line 370, 371, gene names should be written in italics.

Response 17: Thank you for your comments. We have written all gene names in italics. (lines 454, 463)

18.Line 665,669 Please provide the bioinformatics software version used in the manuscript.

Response 18: Thank you for your comments. We have provided versions of all bioinformatics software. (Lines 802, 806, 808, 809, 818, 842, 847, 851, 857, 858)
